# Ubiquitin Ligase *U-Box51* Positively Regulates Drought Stress in Potato (*Solanum tuberosum* L.)

**DOI:** 10.3390/ijms252312961

**Published:** 2024-12-02

**Authors:** Meng Wei, Shantwana Ghimire, Anuja Rijal, Kaitong Wang, Huanhuan Zhang, Huaijun Si, Xun Tang

**Affiliations:** 1College of Life Science and Technology, Gansu Agricultural University, Lanzhou 730070, China; 2State Key Laboratory of Aridland Crop Science, Gansu Agricultural University, Lanzhou 730070, China; 3College of Horticulture, Gansu Agricultural University, Lanzhou 730070, China; 4National Potato Research Program, Nepal Agricultural Research Council, Lalitpur 44700, Nepal; 5College of Agronomy, Gansu Agricultural University, Lanzhou 730070, China

**Keywords:** E3 ubiquitin ligase, *StPUB51*, interacting protein, function analysis

## Abstract

The ubiquitin-proteasome system (UPS) is a key protein degradation pathway in eukaryotes, in which E3 ubiquitin ligases mediate protein ubiquitination, directly or indirectly targeting substrate proteins to regulate various biological processes, including plant growth, hormone signaling, immune responses, and adaptation to abiotic stress. In this study, we identified plant U-box protein 51 in *Solanum tuberosum* (*StPUB51*) as an E3 ubiquitin ligase through transcriptomic analysis, and used it as a candidate gene for gene-function analysis. Quantitative real-time PCR (qRT-PCR) was used to examine *StPUB51* expression across different tissues, and its expression patterns under simulated drought stress induced by polyethylene glycol (PEG 6000) were assessed. Transgenic plants overexpressing *StPUB51* and plants with down-regulated *StPUB51* expression were generated to evaluate drought tolerance. The activities of key antioxidant enzymes-superoxide dismutase (SOD), catalase (CAT), and peroxidase (POD) as well as malondialdehyde (MDA) content in transgenic plants’ leaves were measured under drought conditions. Protein–protein interactions involving StPUB51 were explored via yeast two-hybrid (Y2H) screening, with interaction verification by bimolecular fluorescence complementation (BiFC). StPUB51 was predominantly expressed in stems, with lower expression observed in tubers, and its expression was significantly upregulated in response to 20% PEG-6000 simulated drought. Subcellular localization assays revealed nuclear localization of the StPUB51 protein. Under drought stress, *StPUB51*-overexpressing plants exhibited enhanced SOD, POD, and CAT activities and reduced MDA levels, in contrast to plants with suppressed *StPUB51* expression. Y2H and BiFC analyses identified two interacting proteins, StSKP2A and StGATA1, which may be functionally linked to StPUB51. Collectively, these findings suggest that *StPUB51* plays a positive regulatory role in drought tolerance, enhancing resilience in potato growth and stress adaptation.

## 1. Introduction

Potato (*Solanum tuberosum* L.) is the third largest staple crop in the world, with the advantages of high adaptability, high yield, rich nutrition, and high economic value. Due to its relatively shallow root system, potatoes are very sensitive to a lack of water, especially during the growth and expansion stages of tubers, which can easily lead to potato yield and quality being affected [1]. Therefore, the identification of drought-resistant genes and the study of molecular mechanisms in potatoes provide certain theoretical support for the breeding of drought-resistant and high-yielding potatoes.

The ubiquitin-proteasome system is one of the important and highly selective protein degradation pathways in eukaryotic organisms [2], which plays an active role in plant responses to biotic and abiotic stresses by specifically degrading substrate proteins. The process of ubiquitination involves ubiquitin-activating enzyme E1, ubiquitin-conjugating enzyme E2, and ubiquitin ligase E3, which act in tandem to covalently bind to lysine residues in the substrate proteins in the form of a peptide bond. E3 ubiquitin ligase plays a central role in the precise identification of the target proteins [3], and through the selection of different ubiquitin ligation sites, it regulates the degradation of the target proteins and also affects their activity and localization, which then triggers a series of biological effects. This process is crucial in plant developmental processes, especially in the coping mechanisms of abiotic stresses [4].

U-Box-type ubiquitin ligases (UFD2) were first identified in yeast [5], which is structurally similar to the RING structural domain but differs from the latter in that it lacks conserved cysteine and histidine residues bound to Zn^2 +^ [6]. The U-Box structural domain consists of approximately 70 amino acids and transfers ubiquitin molecules from E2 to target proteins through salt-bridge and hydrogen-bonding interactions to achieve ubiquitin chemical modification. U-Box-type ubiquitin ligases are predominantly found in plants [7], and these proteins act as “regulators” to effectively respond to various environmental stresses [8]. Studies have shown that U-Box E3 ubiquitin ligases are widely involved in biotic/abiotic stresses [9,10,11,12,13], hormones [14,15], growth and development [16,17], autoimmunity [18,19], and sexual reproduction [20,21], suggesting that they are important in the regulation of plant defense mechanisms and adaptation to environmental changes. A total of 64 U-Box structural proteins were identified in *Arabidopsis thaliana*, of which *AtPUB1, AtPUB18, AtPUB19,* and *AtPUB30* were associated with low temperature, drought, and salt stress [22]; 91 U-Box structural proteins were identified in banana (*Musa nana Lour*). Structural proteins, including *MaPUB84* and *MaPUB91,* play a positive regulatory role in response to drought stress [23]. Besides, functional U-Box ubiquitin ligases have also been identified in tomato (*Solanum lycopersicum*), potato (*Solanum tuberosum*), bell pepper (*Capsicum annuum*), and oilseed rape (*Brassica napus*) [24,25].

U-Box E3 ubiquitin ligase has been extensively studied in many other plants, but little is known about the biological function of U-Box in potato. In this study, U-Box ubiquitin ligase *StPUB51* was selected as the target gene by pre-transcriptome analysis to study the biological function of *StPUB51* in potato drought-resistance response. The *StPUB51* (XM_006352603.2) gene was cloned and its expression level and location were analyzed by qRT-PCR and subcellular localization. The transgenic lines were obtained by genetic transformation for drought-resistance analysis. The reliability of the protein interacting with StPUB51 was screened and verified by yeast two-hybridization (Y2H) and BiFC technology, and the regulatory mechanism of *StPUB51* in response to drought stress was preliminically clarified. The experimental results provide a theoretical basis for further study of the *StPUB51* signaling pathway and biological research.

## 2. Results

### 2.1. Bioinformatics Analysis of StPUB51

Bioinformatics studies have shown that *StPUB51* is located on potato chromosome 6, and the CDS region of the total length of 834 bp contains seven introns (Appendix A), and encodes a protein composed of 277 amino acids with a relative molecular weight of 31.82 KDa and a theoretical isoelectric point of 5.59. The chemical formula is C_1414_H_2216_N_390_O_424_S_11_. The StPUB51 protein contains 35 positively charged amino acid residues of Arg and Lys, and 43 negatively charged amino acid residues of Asp and Glu. The secondary and tertiary structure prediction showed that the protein contained α-helix, β-angle, random curl, and extension chains, which accounted for 69.31%, 2.17%, 20.94%, and 7.58%, respectively (Appendix A). The conserved domain of the StPUB51 protein contains a typical U-Box domain and three TPR motifs (Appendix A). According to phylogenetic tree analysis, this protein is most closely related to the wild potato species (KAG5603202.1) (Appendix A). The cis-acting element of 2000 bp is upstream of *StPUB51*. The gene promoter region was found to contain photoresponsive elements, abscisic acid, and cis-acting elements such as cell cycle regulation (Appendix A).

### 2.2. Tissue Specificity of Potato StPUB51 and Expression Analysis Under PEG 6000 Stress

The results of qRT-PCR showed that the expression of *StPUB51* was highest in stems and leaves, and lowest in tubers, and the expression in stems, leaves, and roots was 1.99, 1.97, and 1.29 times higher than that in tubers, respectively (Figure 1A). In order to investigate the possible role of *StPUB51* in adversity signaling, qRT-PCR was used to detect the expression level of *StPUB51* under 20% PEG 6000 stress. The results showed that the expression of this gene in all periods showed an increasing trend compared to 0 h, the lowest expression at 6 h, and the highest expression at 12 h under PEG 6000 treatment (Figure 1B).

### 2.3. Subcellular Localization Assay

Subcellular localization observation showed that EGFP fluorescence signals were present in the nucleus, cytoplasm, and plasma membrane of the control group, while the GFP green fluorescence of the pEGFP-StPUB51 fusion protein in the nucleus of the experimental group was stronger, indicating that the StPUB51 protein was localized in the nucleus (Figure 2).

### 2.4. Genetic Transformation and Characterization of Transgenic Plants StPUB51

The microtuber chip was infected with *Agrobacterium* solution of overexpression vector pRI201-AN-StPUB51 and down-expression vector pCPB121-amiR-StPUB51. The microtuber chip was co-cultured for 2 days (Figure 3A) and transferred to differentiation medium to form callus and differentiated buds (Figure 3B). When the differentiated buds were cut and screened for roots (Figure 3C), plants that were able to take root normally within a week were initially considered transgenic strains. Electrophoretic results showed that the 676 bp fragment of *NPT* II was successfully amplified in both overexpressed and down-expressed lines (Figure 3D). qRT-PCR results showed that the expression levels of StPUB51 in transgenic plants OE-1, OE-2, and OE-3 were 2.09 times, 3.23 times, and 6.80 times those in WT, respectively, which were significantly higher than those in WT. The relative expression levels of RNAi-1, RNAi-2, and RNAi-3 were 0.30 times, 0.35 times, and 0.09 times that of WT, respectively, which were significantly lower than those of WT plants (Figure 3E). These results showed that overexpressed and down-expressed transgenic plants were successfully obtained.

### 2.5. Drought-Resistance Analysis of StPUB51 in Potato

After 14 days of water shortage treatment, it was found that the growth of the OE-*n* strain was significantly better than that of the WT and RNAi-*n* lines, in which the leaves of the RNAi-*n* lines were significantly wilted (Figure 4A). The activities of the antioxidant enzymes and the content of MDA were not significantly different among the plants before treatment. After 14 d of drought stress, the activities of these three antioxidant enzymes were significantly higher in the OE-*n* lines than in the WT and RNAi-*n* lines, and the content of MDA was significantly reduced in the OE-*n* lines compared with that of WT, and the content was significantly increased in the RNAi-n lines (Figure 4B). The results showed that OE-*n* lines had better drought tolerance compared with WT, while the opposite was true for the RNAi-*n* lines.

### 2.6. Yeast Two-Hybrid Identification of StPUB51 Interacting Proteins

Proteins interacting with StPUB51 were screened by the Y2H method, and the results were compared and analyzed (Table 1).

The Y2H experiment was used to verify the interaction between StPUB51, StSKP2A, and StGATA1. Y2H experiment results showed that negative control did not grow in SD-TLHA-x-α-gal defect medium, while both the positive control and experimental group could grow and turn blue. On SDTL-deficient medium, the negative control, positive control, and experimental group all grew, while on SD-TLHA-deficient medium, only the negative control did not grow (Figure 5). It shows that the interaction between the two positive clones is reliable.

### 2.7. BiFC Experiments to Validate StPUB51-Interacting Proteins

YFP was used as a marker in tobacco to characterize the interaction of StPUB51 with StSKP2A and StGATA1. The study found that YFP fluorescence signal expression was found in the nucleus of the experimental group, while no YFP fluorescence signal was detected in the negative control group (Figure 6). These results indicate that StPUB51 interacts with StSKP2A and StGATA1, respectively.

## 3. Discussion

Potatoes face a variety of environmental stresses throughout their life cycle including drought stress which is a particularly complex form of abiotic stress [26]. Drought affects plants through a range of physiological and biochemical responses, impacting morphological structure and metabolic processes. Water deficiency, as a result of drought stress, disrupts normal cell division and proliferation, which can hinder the growth of key organs, including roots, stems, and leaves [27]. Within this context, the U-Box family plays a significant role, with U-Box-type ubiquitin ligases shown to contribute to plant responses to drought and salt stress. These ligases modulate abscisic acid signaling, promote stomatal closure, and regulate drought-responsive transcription factors [28]. In *Arabidopsis thaliana*, ARM/U-Box proteins have been identified as crucial in supporting plant growth and development, as well as in mediating adaptive responses to environmental stresses [18]. *AtPUB12/13* and *AtPUB22/23* improve drought resistance by mediating degradation of ABA receptors ABI1 and PYL9, respectively [29,30].

In order to study whether StPUB51 regulates potato drought stress response by participating in ABA signaling pathway, the physiological function of StPUB51 in potato was further studied. We performed promoter element analysis on the upstream 2000 bp sequence and found several cis-acting elements related to the abiotic stress response, including CCAAT-box, ABER, and GATA motifs. The CCAAT-box serves as a binding site for MYB family transcription factors, which play a role in drought stress responses [31,32,33]. ABER cis-acting elements are involved in the regulation of drought responses and ABA signaling pathways [34], while GATA transcription factors contribute to the regulation of drought, cold, and salt stresses, among other stress responses [35,36]. These findings suggest that *StPUB51* likely plays a crucial role in enhancing potato stress tolerance [37]. Subcellular localization analysis further showed that *StPUB51* protein localizes within the nucleus which suggest its role in genetic and metabolic regulation. qRT-PCR analysis revealed differential expression of *StPUB51* across various potato tissues, with the highest expression in stems and leaves and the lowest in tubers. Notably, *StPUB51* expression increased after treatment with 20% PEG 6000, indicating its potential role in drought tolerance.

Reactive oxygen species (ROS) are oxygen-containing reactive compounds generated during cellular metabolism, primarily comprising superoxide (O_2_^−^) and hydrogen peroxide (H_2_O_2_) [38]. Environmental stresses such as drought, salt, and osmotic stress can trigger excessive ROS production in plants, leading to cellular damage and potential cell death [39]. To counteract oxidative stress and maintain cellular redox balance, plants enhance the activity of antioxidant enzymes, including superoxide dismutase (SOD), catalase (CAT), and peroxidase (POD), which act as key protective mechanisms [40,41]. Malondialdehyde (MDA), an indicator of oxidative damage to cell membranes, reflects the extent of cellular membrane permeability damage [42]. The *StPUB51* promoter region contains ABRE cis-acting elements involved in drought response. To investigate the role of *StPUB51* in potato drought stress, this study generated potato lines with *StPUB51* overexpression and RNA interference (RNAi) through genetic transformation. Under drought conditions, *StPUB51* overexpression lines displayed increased activities of antioxidant enzymes (POD, CAT, SOD), reduced MDA content, enhanced ROS-scavenging capacity, and improved drought resistance compared to wild-type (WT) plants. In contrast, RNAi lines showed decreased antioxidant activity, increased MDA content, and reduced drought tolerance, suggesting that *StPUB51* overexpression minimizes membrane lipid peroxidation, shielding potato plants from ROS toxicity under drought stress. Similar oxidative damage patterns, including elevated ROS levels, have been observed under drought stress in *Arabidopsis* [43] and maize [44], where SOD catalyzes peroxide production to H_2_O_2_, while CAT, APX, and POD help decompose H_2_O_2_, mitigating cellular damage.

Screening for interacting proteins is crucial for understanding protein functions and cell signaling, as it reveals complex interaction networks among proteins, providing key insights into molecular mechanisms underlying plant cellular processes [45]. In this study, yeast two-hybrid (Y2H) and bimolecular fluorescence complementation (BiFC) assays were used to validate interactions, demonstrating that StPUB51 interacts with StSKP2A and StGATA1. This suggests that the potato *StPUB51* may participate in multiple signaling pathways. Through the function analysis of F-box and GATA, it was found that proteins containing F-box domain, such as StSKP2A, can specifically recognize and bind substrates, and play a role in protein degradation, receptor recognition, and signal regulation [46,47]. Overexpression of F-Box Of Flowering 2(FOF2) increases *Arabidopsis* tolerance to drought and regulates the ABA-mediated stress response [48]. At the same time, GATA transcription factors like StGATA1 can bind to WGATAR sequences in target gene promoters to activate or repress transcription, influencing cellular growth, carbon and nitrogen metabolism, flower development, and chloroplast biogenesis [49,50]. Further analysis and verification of StSKP2A and StGATA1 functions will help clarify the specific signaling pathways through which *StPUB51* mediates stress responses in potato.

## 4. Materials and Methods

### 4.1. Plant Materials and Growth Conditions

The potato variety “Desiree” shows sensitivity to drought. We chose it as the experimental material. Stem segments of potato variety “Desiree” were inoculated in MS solid medium containing 3% and 6% sucrose, and placed in a culture environment at (22 ± 1) °C, light intensity of 2000 Lx, and 16 h of light/8 h of darkness. After 30 d, the plants inoculated with 6% sucrose MS solid medium were transferred to a dark environment, and cultured until miniature potatoes were obtained [51]. At the same time, potato plants in 3% sucrose MS solid medium after 30 d of incubation were transplanted into 10 cm × 10 cm (vermiculite: nutrient soil = 2:1) pots under the same conditions as above, and after 30 d of incubation, the plants with essentially the same growth conditions were selected for subsequent stress treatments. Once the tuberization stage was completed, roots, stems, leaves, and tubers were collected for tissue-specific analysis. When the plants reached a height of about 25 cm, they were irrigated with 20% PEG 6000. The leaves were collected at 0 h, 3 h, 6 h, 12 h, and 24 h, and stored at −80 °C. The samples (plant number, root number, stem/stem node number, leaf number, tuber number) were collected for three biological replicates.

### 4.2. Cloning of StPUB51

In this study, *StPUB51* was used to analyze the drought-resistance function of potatoes. The CDS sequences of genes are shown in Appendix A. The *StPUB51* sequence was retrieved from the potato database Spud DB (http:spuddb.uga.edu/index.shtml, accessed on 13 June 2024). Total RNA extraction and cDNA first-strand synthesis of potato cultivar “Desiree” were performed using the TRNzol Universal Plant Total RNA Extraction Kit and the FastKing gDNA Dispelling RT Super Mix Reverse Transcription Kit, respectively. The CDS region of the *StPUB51* was cloned using cDNA from the leaves of the potato variety “Desiree” as a template. The primer sequences are shown in Appendix A. The reaction system was prex TaqTM (TaKaRa TaqTM Version 2.0 + dye) 10 μL, ddH_2_O 6.5 μL, cDNA template 1.5 μL, and specific primer F/R 1 μL. The reaction conditions were as follows: predenaturation at 95 °C for 3 min, denaturation at 94 °C for 25 s, annealing at 60 °C for 25 s, and extension at 72C for 1 min, for a total of 34 min. The reaction conditions were as follows: pre-denaturation at 95C for 3 min, denaturation at 94 °C for 25 s, annealing at 60 °C for 25 s, extension at 72 °C for 1 min, a total of 34 cycles, and final extension at 72 °C for 5 min. The clone of StPUB51 is shown in Appendix A.

### 4.3. Bioinformatics and Expression Analysis of StPUB51

The sequence of potato *StPUB51* was obtained by the NCBI method. The basic physicochemical properties were analyzed by ProtParam, SOPMA, SWISS-MODEL, and SMART. Main page was used to predict the secondary and tertiary structures and conserved domains of proteins. By comparing the amino acid sequences of BLAST and StPUB51, homologous amino acid sequences of 11 species were obtained. DNAMAN6.0 and MEGA7.0 software were used to analyze the protein sequences, and phylogenetic trees of homologous proteins were constructed. Cis-acting element analysis was performed using PlantCARE.

### 4.4. Analysis of StPUB51 Expression by qRT-PCR

The primers are specific for *StPUB51*; they were designed by the online website Primer 3 Plus and the differences in the expression of the *StPUB51* in the roots, stems, leaves, and tubers of the potato variety “Desiree” were statistically analyzed, with the potato *StEFlα* (GenBank No: AB061263.1) used as an internal reference gene. The relative expression of StPUB51 in various tissues was analyzed by qRT-PCR, and the expression of StPUB51 in potato at different time periods under 20% PEG stress was analyzed by qRT-PCR. The experiment was repeated with three techniques. The primer sequence was shown in Appendix A. The reaction system was 1 μL cDNA (100 ng) template, 10 μL 2 × Universal Blue SYBR Green qPCR Master Mix, 1 μL qRT-StPUB51-F/R, and 7.0 μL ddH_2_O. The reaction conditions were as follows: predenaturation at 95 °C for 30 s, denaturation at 95 °C for 15 s, and annealing at 60 °C for 30 s for a total of 40 cycles. The relative expression of *StPUB51* was calculated using the method of 2^−ΔΔCt^ [52].

### 4.5. Subcellular Localization of StPUB51 Protein

The subcellular location of StPUB51 protein was predicted by the online website PSORT (https://www.genscript.com/psort.html accessed on 25 April 2024), and the prediction results showed that StPUB51 protein was localized in the nucleus, in order to further verify the reliability of the prediction results. The *StPUB51* subcellular localization vector was constructed by homologous recombination, and specific prim.ers (Appendix A) were designed according to the sequences of the *StPUB51* and the pCAMBIA1300-35S-EGFP vector, and a cDNA library of potato leaves from the potato variety “Desiree” was used as a template to amplify the CDS sequence of the *StPUB51* without the terminator. The PCR product with the appropriate amplified bands was inserted into the *Kpn* I and *Xba* I double-cleaved pCAMBIA1300-35S-EGFP linearized vector and introduced into *E. coli* receptor DH5α; the specific operation was referred to the method of Wang [50]. The recombinant plasmid was named pEGFP-StPUB51 after the successful sequencing and comparison, and then the empty plasmid was inserted into pCAMBIA1300-35S-EGFP linearized vector and imported into *E. coli* receptor DH5α; the specific operation was referred to the method of Wang [53]. The empty plasmid pCAMBIA1300-35S-EGFP and the recombinant plasmid pEGFP-StPUB51 were then introduced into *Agrobacterium rhizogenes* GV3101, respectively. Referring to the method of Qi et al. [54], the infiltration solution was injected from the backside of the tobacco leaf (the 2nd–4th leaves from top to bottom of 5–7-week-old tobacco leaves) with a disposable sterile injector needle and the infiltration area was marked. After incubation under dark conditions for 1 d and then transferred to light culture for 2 d, the distribution of green fluorescent signals at the excitation wavelength of 488 nm was observed under a laser confocal scanning electron microscope to determine the expression sites of StPUB51 protein in the cells.

### 4.6. Construction of Plant Expression Vectors

The *StPUB51* overexpression vector was constructed using homologous recombination, and specific primers were designed based on the StPUB51 sequence and the vector pRI201-AN sequence (Appendix A). The PCR gel recovery product was inserted into the pR1201-AN vector that was double-enzymatically cleaved by *Nde* I and *Sal* I. The pR1201-AN vector was used for the sequencing of StPUB51 and the pRI201-AN vector. The recombinant plasmid pRI201-AN-StPUB51 was obtained as verified by double digestion and identified by sequencing. For the down-regulated expression vectors, precursor primers (I, II, III, and IV) were designed using Oligo from the online website WMD3 (Appendix A), and target fragments were obtained by standard PCR [55]. The PCR product was ligated into the pMD18-T vector, and the recombinant plasmid T18-StPUB51 was obtained after double digestion verification and sequencing. The pCPB121 vector was digested with T18-StPUB51 recombinant plasmid using restriction endonuclease *Kpn* I and *Sac* I. The gel was recovered for PCR product, and the precursor small fragment was ligated with a large fragment of linearized vector using T4 ligase and was compared with the sequenced product. The recombinant plasmid pCPB121-amiR-StPUB51 was obtained by sequencing and comparison. pCPB121-amiR-StPUB51 was transformed into *Agrobacterium rhizogenes* GV3101 with the successfully identified recombinant plasmid [56].

### 4.7. Genetic Transformation of Potato StPUB51

The genetic transformation of microtuber was referred to the method of Si et al. [57]. The potato variety “Desiree” microtuber was used as the test material, and the microtuber was cut into 0.3–0.4 cm slices with a sterile blade. The potato slices were transferred to *Agrobacterium tumefaciens* solution containing recombinant plasmids pRI201-AN-StPUB51 and pCPB121-amiR-StPUB51, and then infiltrated for 7–8 min. The residual bacterial solution on the potato slices was sucked up with sterile dry filter paper and spread on the co-culture medium, and then the potato slices were incubated in the dark for 2 d at 28 °C. The medium was changed once a week. When the differentiated shoots grew to about 1.5 cm, they were cut and transferred to the rooting medium containing Kan (50 mg/mL) and Cef (100 mg/mL) for three rooting screenings, and the normal rooting was initially identified as transgenic plants in about one week. To further identify the transgenic plants, genomic DNA of WT plants and transgenic plants was extracted by CTAB, and WT were used as negative control, and recombinant plasmids pRI201-AN-StPUB51 and pCPB121-amiR-StPUB51 were used as positive control, respectively. Electrophoresis results showed that the transgenic plants all successfully amplified a 676 bp *NPT* II fragment. The primer sequences are shown in Appendix A. RNA from the transgenic plants and WT plants was extracted and reverse transcribed into cDNA, and the relative expression of the *StPUB51* in the transgenic plants was detected by qRT-PCR using the WT plants as a control. pRI201-AN-StPUB51 and pCPB121-amiR-StPUB51 were named as transgenic plants, and the transgenic plants were named as pCPB121-amiR-StPUB51 and pCPB121-amiR-StPUB51, respectively. The transgenic plants were named OE-*n* and RNAi-*n*.

### 4.8. Drought-Stress Treatment of Transgenic Plants

The stem segments of transgenic potato variety “Desiree” were inoculated in MS solid medium containing 3% sucrose and placed in a light culture greenhouse under the same conditions as in 4.1. After the plants had grown to 30 d, they were planted into pots of 10 cm×10 cm (vermiculite: nutrient soil = 2:1) and maintained at a temperature of (22 ± 1) °C and light intensity of 2000 Lx, with 16 h of light/8 h of darkness for about 30 d. Plants with essentially the same growth conditions of the treatments were selected, and after 14 d of water deficit and drought treatment, the top third to fifth leaves of the plants were collected and quickly placed in liquid nitrogen for freezing and preservation. The activities of three key antioxidant enzymes (SOD, POD, and CAT) and the content of MDA were determined. SOD activity was determined by the method described by Giannopolitis and Ries [58]; POD activity was determined by the method described by Maehly [59]; and CAT activity was determined by the method described by Aebi [60]; MDA content was determined by the thiobarbituric acid (TBA) method described by Heath and Packer [61]. In the above experiments, three plants were selected from each line, and each plant was subjected to three biological replicates and three technical replicates.

### 4.9. Yeast Two-Hybrid Experiment

To further study the function of StPUB51 in potato drought stress response, the pGBKT7-StPUB51 recombinant vector was constructed by homologous recombination using *Nde* I and *Sal* I as restriction endonucleases. The primer sequences are shown in Appendix A. Under the premise of excluding the pGBKT7-StPUB51 bait vector from toxicity and self-activating activity, the potato cDNA library was co-transfected with pGBKT7-StPUB51 in *yeast* cells AH109 by the PEG/LiAc-mediated method. After incubation for 3–5 d at 30 on SD-TL selection medium, a colony of about 2 mm was picked and put into 20 μL of 0.9% NaCl solution for dilution, aspirate 4 μL spot in SD-TLHA-X-α-gal, and incubate inverted at 30 °C for 4–5 d. Then, blue colonies were picked to be sent for colony PCR, and sequencing results were analyzed by sequencing [62]. We mainly analyzed the following two proteins that may be involved in the regulation of plant drought stress response: F-box and GATA. They were named StSKP2A (XM_015312788.1) and StGATA1 (XM_006355679.2), respectively, according to their functions in *Arabidopsis*. Gene CDS sequences are shown in Appendix A. The cloning of *StSKP2A* and *StGATA1* is shown in Appendix A.

### 4.10. Yeast Rotary Validation Analysis

*Nde* I and *Bam*H I were used as restriction endonucleases to construct pGADT7-StSKP2A and pGADT7-StGATA1 recombinant vectors by homologous recombination. The primer sequences are shown in Appendix A. pGADT7-StSKP2A and pGADT7-StGATA1 were co-transfected into *yeast* cells AH109, and co-transfected into SD-TL medium for 3–5 d at 30 °C. Then, 4 μL of spot samples were taken and diluted in 20 μL of 0.9% NaCl and SD-TL medium for 3–5 d at 30 °C. After this, colonies of about 2 mm were picked and put into 20 μL of 0.9% NaCl solution for dilution, aspirate. Further, 4 μL spot samples in SD-TLHA-X-α-gal were incubated at 30 °C for 4–5 d; blue spotted colonies were picked and put in 0.9% NaCl solution, and diluted in gradients of 10^0^, 10^−1^, 10^−2^, and 10^−3^, respectively. Spot sample in SD-TL, SD-TLHA, and SD-TLHA-X-α-gal, respectively, were incubated at 30 °C for 5 d. The sample was then incubated for 4~5 d [63,64].

### 4.11. Complementary Analysis of Bimolecular Fluorescence

To verify the reliability of the Y2H result, the terminator-free *StPUB51*, *StSKP2A*, and *StGATA1* were amplified by PCR. The primer sequences are shown in Appendix A, and pSPYCE-StPUB51, pSPYNE-StSKP2A, and pSPYNE-StGATA1 recombinant vectors were constructed by homologous recombination. pSPYCE-StPUB51, pSPYNE-StSKP2A, and pSPYNE-StGATA1 recombinant vectors were used, and the empty and recombinant plasmids were transformed into *Agrobacterium tumefaciens* GV3101, respectively. Primer sequences are shown in Appendix A. The bacterial fluids of pSPYCE-StPUB51 and pSPYNE-35S were mixed with pSPYNE-StSKP2A/StGATA1 in a 1:1 ratio and injected into tobacco leaves using pSPYCE-StPUB51 and pSPYNE-35S as a blank control. Injection and infiltration were performed concerning subcellular localization, and a yellow color was observed at the excitation wavelength of 514 nm by laser confocal scanning electron microscopy. Fluorescent protein expression site at 514 nm excitation wavelength [65].

## 5. Conclusions

In this study, we cloned the *StPUB51* from potato and analyzed its expression across different tissues of the cultivar “Desiree” using qRT-PCR. *StPUB51* showed the highest expression in stems, and its expression was notably induced by PEG-induced stress. Subcellular localization confirmed that the *StPUB51* protein is localized within the nucleus. Under drought stress, transgenic plants overexpressing *StPUB51* exhibited increased activities of antioxidant enzymes (SOD, POD, and CAT), enhanced ROS-scavenging capacity, improved drought tolerance, reduced MDA content, and decreased cellular damage compared to wild-type plants. Conversely, down-regulation of *StPUB51* resulted in diminished drought resilience, underscoring its role in drought tolerance. Furthermore, the interactions between StPUB51 and proteins StSKP2A and StGATA1 were verified using yeast two-hybrid (Y2H) and bimolecular fluorescence complementation (BiFC) assays. These findings offer a theoretical foundation for understanding the mechanisms and signaling pathways involved in potato responses to drought stress.

## Figures and Tables

**Figure 1 ijms-25-12961-f001:**
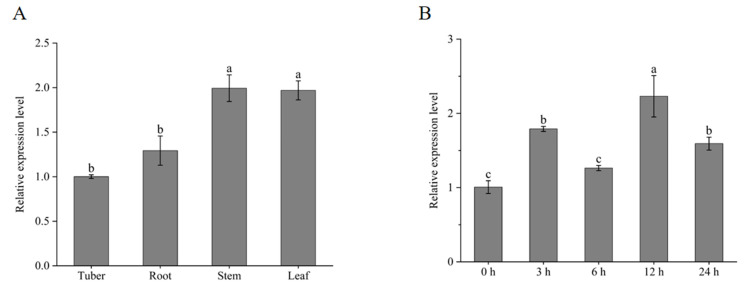
Tissue-specific and drought-mimicking expression pattern analysis of potato *StPUB51*. (**A**) Relative expression of *StPUB51* in different organs of “Desiree” potato; (**B**) 20% PEG 6000 simulated drought-stress treatment. Note: A one-way ANOVA was used in this experiment and the error bars represent the standard errors (*n* = 3). Different small letters mean significant differences (*p* < 0.05).

**Figure 2 ijms-25-12961-f002:**
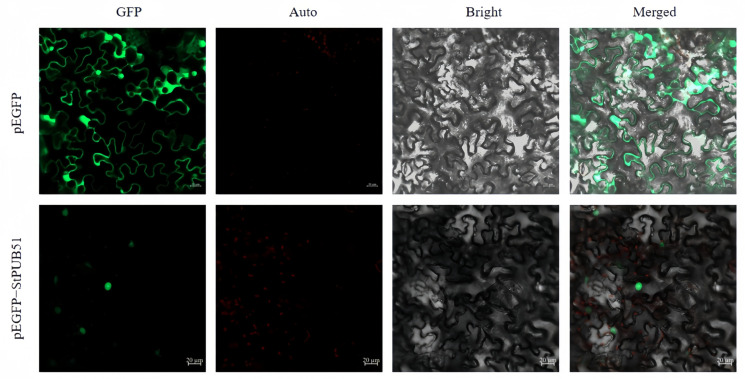
Subcellular localization of StPUB51 protein. Transient expression of pCAMBIA1300-35S-EGFP and pEGFP-StPUB51 fusion proteins in tobacco; GFP: green fluorescence; Auto: red chloroplast autofluorescence; Bright: bright field; Merge: superimposed field; green color was GFP fluorescence; red color was chloroplast autofluorescence (bar = 20 μm).

**Figure 3 ijms-25-12961-f003:**
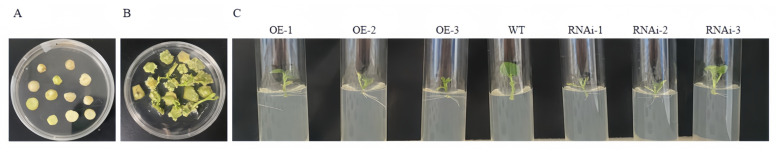
Acquisition and characterization of transgenic plants. (**A**) Potato chip co-culture; (**B**) healing group differentiation of shoots; (**C**) rooting screening of transgenic plants; (**D**) PCR assay of transgenic plants; (**E**) relative expression assay of transgenic plants. Note: Different small letters mean significant differences (*p* < 0.05, *n* = 3).

**Figure 4 ijms-25-12961-f004:**
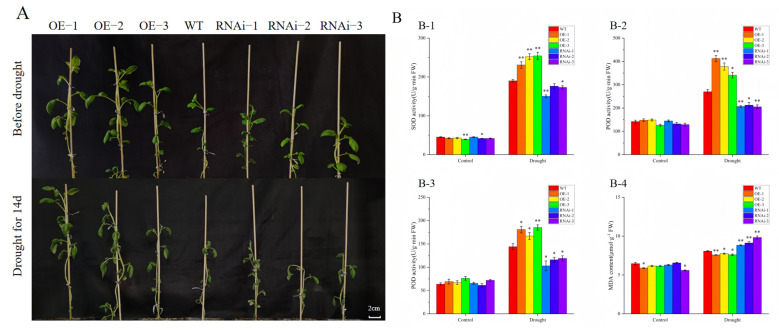
Physiological and biochemical indexes measured in transgenic plants under natural drought conditions. (**A**) Phenotypic changes of transgenic potato under drought stress for 14 d; scale bar = 2 cm; (**B**) (**B-1**) SOD activity; (**B-2**) POD activity; (**B-3**) CAT activity; (**B-4**) MDA activity. Note: Each column represents the mean ± SE (*n* = 3; * *p* < 0.05; ** *p* < 0.01).

**Figure 5 ijms-25-12961-f005:**
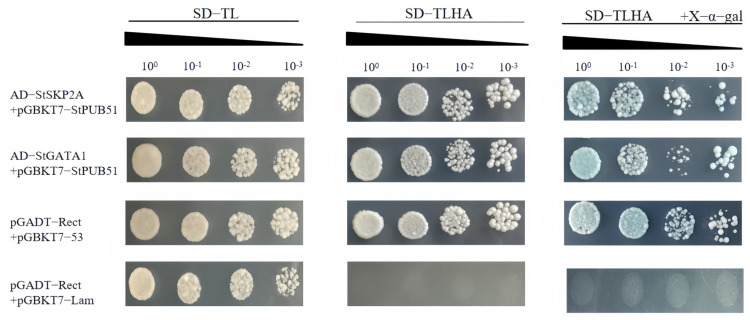
Y2H experiment to verify the interaction between StPUB51 and StSKP2A and StGATA1. Note: Positive control: pGADT-Rect + pGBKT7-53; Negative control: pGADT-Rect + pGBKT7-Lam; the rest are experimental groups.

**Figure 6 ijms-25-12961-f006:**
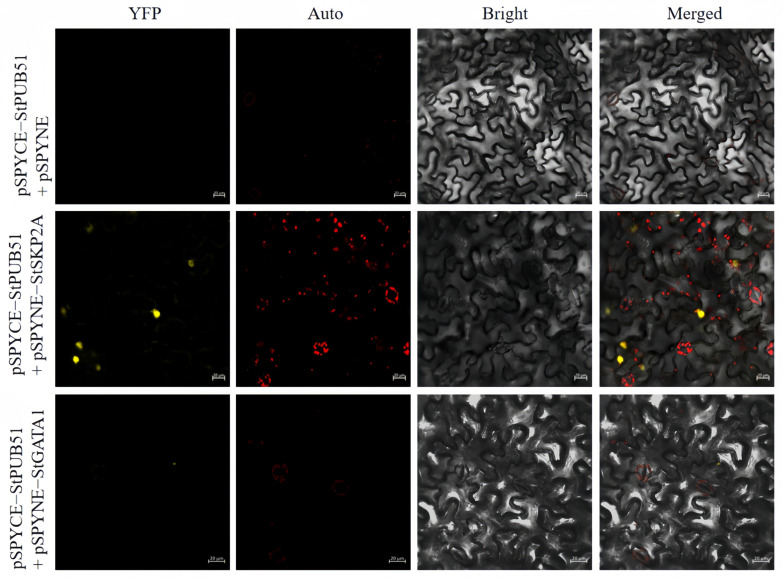
BiFC experiments verified the interaction of StPUB51 with StSKP2A and StGATA1. The pSPYCE-StPUB51 + pSPYNE-StSKP2A/StGATA1 was used as the experimental group, and pSPYCE-StPUB51 + pSPYNE was used as the negative control. YFP: yellow fluorescence; Auto: red chloroplast autofluorescence; Bright: bright field; Merge: superimposed field; yellow color was YFP fluorescence; red color was chloroplast autofluorescence (bar = 20 μm).

**Table 1 ijms-25-12961-t001:** The information of StPUB51-interacting proteins.

NCBI Accession No.	PGSC ID	Annotations	Subcellular Localization
XM_015312788.1	Soltu.DM.04G025280.1	F-box/RNI-like superfamily protein	Nucleus
XM_006355679.2	Soltu.DM.05G023550.1	GATA transcription factor	Nucleus
XM_006346493	Soltu.DM.07G011150.1	cytidine deaminase	Nucleus
XM_006360024	Soltu.DM.07G025540.1	polyubiquitin	Nucleus
XM_006350714	Soltu.DM.06G031560.1	PGR5-like protein 1B, chloroplastic	Chloroplast
XM_006345193.2	Soltu.DM.12G026280.1	oligouridylate binding protein 1B	Chloroplast
XM_006349677.2	Soltu.DM.09G020540.1	Primosome PriB/single-strand DNA-binding	Nucleus
XM_006339612.2	Soltu.DM.01G034630.1	Mitochondrial substrate carrier family protein	Mitochondrion
XM_006358242.2	Soltu.DM.07G003310.1	Raffinose synthase family protein	Chloroplast, Cytoplasm
XM_006355002	Soltu.DM.04G003360.1	2-oxoglutarate (2OG) and Fe(II)-dependent oxygenase superfamily protein	Cytoplasm
XM_006350817	Soltu.DM.06G032790.1	PAP/OAS1 substrate-binding domain superfamily	Chloroplast, Nucleus
XM_006359236.2	Soltu.DM.04G005650.1	conserved hypothetical protein	Cell membrane, Chloroplast, Nucleus

## Data Availability

Data are contained within the article and Appendix A.

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
