# Peer review of "Ubiquitin Ligase U-Box51 Positively Regulates Drought Stress in Potato (Solanum tuberosum L.)"

_ijms, 2024, doi:10.3390/ijms252312961_

Round 1
Reviewer 1 Report
Comments and Suggestions for Authors
The paper describes results of a laboratory research, experiment on the examination of the biological function of ubiquitin ligase U-Box51 in regulating drought stress in potato. The topic is interesting and the results could provide progress in current knowledge on the regulation of drought stress in potato. The manuscript can be considered for publication in the International Journal of Molecular Sciences after revision have been made.
I suggest to change some keywords – "potato" and "drought stress" are the repetition of the title words. Please find words that more detail reflecting the scientific content of the paper.
The aim of the study and the hypothesis put forward in the study should be clearly stated.
Section Materials and Methods should be completed. The sample size of plant material used in each stage of the experiment should be added (number of plants, number of roots, number of stems/stem segments, number of leaves, number of tubers). The statistical methods used should be described. Why the potato variety ‘Desiree’ was used? The drought sensitivity of this variety should be added. Why were tobacco leaves used to analyse bimolecular fluorescence?
Lines 382-383: "The medium was changed once a week. The medium was changed once w week" – please remove repetition.
The Authors mixes results with methods. Lines 84-86, 102-104, 119-127, 137-146, 148-149, 164-165, 167-169, 183-185, 187-194, 207-212 should be moved to the Materials and Methods section.
The Authors should better discuss the significance of their findings.
Author Response
Dear Editor and Reviewers,
We would like to express our sincere gratitude for your valuable time and constructive feedback on our manuscript. Your insightful comments have been instrumental in improving the quality of our work, and we have carefully addressed each point raised. In this response, we provide a detailed, point-by-point reply to the comments and concerns, along with the corresponding revisions made to the manuscript. Additionally, we have included a revised version of the manuscript and a marked-up version highlighting the specific changes for your convenience.
We hope that these revisions sufficiently address your concerns and enhance the quality of our manuscript, making it suitable for publication in International Journal of Molecular Sciences. Please do not hesitate to reach out if further clarifications or additional revisions are required.
Thank you again for your thoughtful and constructive input.
Sincerely,
The Authors
The paper describes results of a laboratory research, experiment on the examination of the biological function of ubiquitin ligase U-Box51 in regulating drought stress in potato. The topic is interesting and the results could provide progress in current knowledge on the regulation of drought stress in potato. The manuscript can be considered for publication in the International Journal of Molecular Sciences after revision have been made.
Response: Thank you for your recognition of the quality of the manuscript and our work.
1.I suggest to change some keywords – "potato" and "drought stress" are the repetition of the title words. Please find words that more detail reflecting the scientific content of the paper.
Response: Thanks for the constructive comment and time. We removed keywords that duplicate the title and added two new keywords, "E3 ubiquitin ligas" and "gene function analysis".
2.The aim of the study and the hypothesis put forward in the study should be clearly stated.
Response: We are thankful for the critical review. We have rewritten the last paragraph of Introduction to make the research purpose clearer. The discussion section has also been modified, and the hypotheses and results proposed by the experiment have become clearer.
- Section Materials and Methods should be completed. The sample size of plant material used in each stage of the experiment should be added (number of plants, number of roots, number of stems/stem segments, number of leaves, number of tubers).
Response: We are thankful for the critical review. The sample collection quantity (number of plants, number of roots, number of stems/stem segments, number of leaves, number of tubers) was subjected to 3 biological replicates, and 3 technical replicates were performed in the qRT-PCR experiment, which was supplemented to materials and methods 4.1 and 4.4.
- The statistical methods used should be described.
Response: We are thankful for the critical review. The statistical data were expressed as three repeated mean ± standard deviation, a one-way analysis of variance adjusted by Duncan's (p < 0.05), as indicated in the notes in Figure 1.
5.Why the potato variety ‘Desiree’ was used? The drought sensitivity of this variety should be added.
Response: We are thankful for the critical review. “Physiological and molecular responses to drought stress in Desiree potato plants” found that the relative water content of leaves of "Desiree" decreased and the content of malondialdehyde increased under drought stress. The activity of antioxidant enzymes and the expression of some drought response genes were also changed, indicating that they were sensitive to drought stress. Based on our experimental results, we describe the drought sensitivity of "Desiree" at methods 4.1.
- Why were tobacco leaves used to analyse bimolecular fluorescence?
Response: We are thankful for the critical review. Tobacco has a relatively simple and mature genetic transformation system, short growth cycle and large leaf tissue, and it is easy to observe the fluorescence signal generated by the interaction of target proteins under the microscope. See “Visualization of protein interactions in living plant cells using bimolecular fluorescence complementation”.
- Lines 382-383: "The medium was changed once a week. The medium was changed once w week" – please remove repetition.
Response: We are thankful for the critical review. We are sorry for the carelessness in the writing and we have revised it.
- The Authors mixes results with methods. Lines 84-86, 102-104, 119-127, 137-146, 148-149, 164-165, 167-169, 183-185, 187-194, 207-212 should be moved to the Materials and Methods section.
Response: We are thankful for the critical review. Based on your valuable input, we have rewritten the results and methods to make them clearer.
- The Authors should better discuss the significance of their findings.
Response: We are thankful for the critical review. We have rewritten the discussion section and highlighted the experimental findings. In addition, we have added three references for broader comparison and discussion.
Reviewer 2 Report
Comments and Suggestions for Authors
The MS studied the function of one E3 ubiquitin ligase gene in drought tolerance in potato. The MS was well written and carefully presented. I have a few questions or improvement suggestions:
1) in abstract, when StPUB51 first appeared, it should have full name.
2) Line 45, species (Table 1)?
3) Line 73, “Little is known about the biological function of U-box, a member of E3 ubiquitin ligases.” As I know, U-box E3 ubiquitin ligases were broadly investigated in many other plants. This sentence is debatable.
4) In Figure 1, the relative expression of StPUB51 under drought stress was slightly induced, only twofold. The author mentioned that this gene was selected from transcriptome data, so it should be one significant-induced gene. So, why choose this gene? How about the expression induction?
5) In Figure 2, the chloroplast autofluorescence was almost invisible.
6) In Figure 4, The transgenic plants before drought stress showed significantly different phenotypes, with plant height of OE were higher. The growth situation has impact on the drought tolerance. So, the drought resistance of OE could be due to the better growth. How about the author’s consideration? And, for the POD, SOD, CAT, MDA measurement, the author should compare the difference between WT and transgenic lines, not just compare between normal and drought.
Author Response
Dear Editor and Reviewers,
We would like to express our sincere gratitude for your valuable time and constructive feedback on our manuscript. Your insightful comments have been instrumental in improving the quality of our work, and we have carefully addressed each point raised. In this response, we provide a detailed, point-by-point reply to the comments and concerns, along with the corresponding revisions made to the manuscript. Additionally, we have included a revised version of the manuscript and a marked-up version highlighting the specific changes for your convenience.
We hope that these revisions sufficiently address your concerns and enhance the quality of our manuscript, making it suitable for publication in International Journal of Molecular Sciences. Please do not hesitate to reach out if further clarifications or additional revisions are required.
Thank you again for your thoughtful and constructive input.
Sincerely,
The Authors
The MS studied the function of one E3 ubiquitin ligase gene in drought tolerance in potato. The MS was well written and carefully presented. I have a few questions or improvement suggestions:
Response: Thank you for your recognition of the quality of the manuscript and our work.
- in abstract, when StPUB51 first appeared, it should have full name.
Response: StPUB51 stands for plant U-box protein 51 in Solanum tuberosum. We described StPUB51 when it first appeared in the abstract.
- Line 45, species (Table 1)?
Response: We are sorry for the carelessness in the writing and we have revised “Table 1”.
- Line 73,“Little is known about the biological function of U-box, a member of E3 ubiquitin ligases.”As I know, U-box E3 ubiquitin ligases were broadly investigated in many other plants. This sentence is debatable.
Response: We are thankful for the critical review. We redescribed it. We redescribed it and changed it to “Little is known about the biological function of U-box, a member of E3 ubiquitin ligases. Be changed to U-box E3 ubiquitin ligase has been extensively studied in many other plants, but little is known about the biological function of U-box in potato.”
- In Figure 1, the relative expression of StPUB51 under drought stress was slightly induced, only twofold. The author mentioned that this gene was selected from transcriptome data, so it should be one significant-induced gene. So, why choose this gene? How about the expression induction?
Response: We appreciate the critical and insightful feedback. U-box E3 ubiquitin ligase, as a key component of the ubiquitin modification system, plays a crucial role in maintaining the stability of target proteins. Unlike functional proteins, post-translational regulatory factors such as E3 ubiquitin ligases typically exhibit lower expression levels. However, even a modest upregulation in their expression can significantly influence the degradation of target proteins, thereby inducing substantial changes in cellular metabolism. This underscores the importance of StPUB51 despite its relatively low induction level under drought stress.
- In Figure 2, the chloroplast autofluorescence was almost invisible.
Response: We are thankful for the critical review. We adjusted the degree of background to make chloroplast autofluorescence appear more pronounced.
- In Figure 4, The transgenic plants before drought stress showed significantly different phenotypes, with plant height of OE were higher. The growth situation has impact on the drought tolerance. So, the drought resistance of OE could be due to the better growth. How about the author’s consideration?
Response: We are thankful for the critical review. In this experiment, the same cultivation method was adopted, and the OE lines showed faster growth rate, so the plant height was higher than that of the WT and RNAi groups, but these differences did not reach a significant level.
And, for the POD, SOD, CAT, MDA measurement, the author should compare the difference between WT and transgenic lines, not just compare between normal and drought.
Response: We are thankful for the critical review. The comparison of WT and transgenic lines has been made in the manuscript figure 4B. We changed the horizontal title of Figure 4B from "normal" to "control" to show the true meaning.